# V-Former: Offline RL with Temporally-Extended Actions

## Abstract

In this paper, we propose an offline reinforcement learning (RL) method that learns to take temporally extended actions, can handle narrow data distributions such as those produced by mixtures of multi-task demonstrations, and can train on data with different control frequencies. This combination of properties makes our proposed method especially well-suited for robotic offline RL, where datasets might consist of (narrow) demonstration data mixed with (broader) suboptimal data, and control frequencies can present a particularly significant challenge. We show that offline RL with temporally extended "action chunks" can be performed efficiently by extending the implicit Q-learning (IQL) approach, in combination with expressive Transformer-based policies for representing temporally extended open-loop action sequences. Our experiments show that our method both improves over prior approaches on simulated robotic demonstration data and outperforms prior works that aim to learn from data at multiple frequencies.

## 1 Introduction

Offline reinforcement learning (RL) has the potential to enable highly capable policies to be learned from suboptimal data, utilizing large previously collected datasets, and making RL practical for settings where online exploration is difficult or expensive (Levine et al., 2020). However, while in principle offline RL combines the best parts of reinforcement learning (reward maximization) and imitation learning (using offline data), in practice it also combines the challenges from both domains. In particular, the nature of the data distribution has a large influence on the difficulty of the offline RL problem, and some data distributions present especially significant challenges: data that is "narrow" in the sense of capturing near-deterministic behaviors, but "suboptimal" in the sense that these behaviors might be suboptimal, non-Markovian, and multi-modal is especially tough to handle, and such data distributions are common when learning from multiple demonstrated behaviors that are too suboptimal for direct imitation, but do not contain the kind of high-coverage noise that is often assumed in RL theory (Agarwal et al., 2019). In such settings, effective offline RL methods need to essentially combine *both* strong imitation learning techniques and robust and stable reinforcement learning procedures.

In this paper, we argue that *temporally extended actions* can provide a more effective and performant method for handling offline RL with a mixture of narrow and broad data distributions, providing an especially effective method for the commonly encountered case in robotics where the data is produced by a mixture of different behaviors. The intuition behind this idea is that non-Markovian, multimodal demonstration data might have sparse "decision points" where the RL algorithm should switch from one behavior to another, and in-between these points the learned policy should largely imitate the data, because the narrow data distribution does not actually support inference of counterfactual actions outside of a few key "decision points." Hence, the RL algorithm should make decisions more sparsely in time than every time step. We instantiate this idea by using sequence models (*i.e.*, Transformers) to model temporally extended actions. Inspired by temporally extended actions and *action chunking* in imitation learning (Zhao et al., 2023; Chi et al., 2023), we formulate an offline RL method that trains policies and value functions that handle multiple time steps at a time, such that the policy outputs a chunk of actions for multiple time steps into the future that are then executed in sequence. While this approach has been found to be highly effective in recent imitation learning work, extending this idea into offline RL presents significant challenges, as it also requires a Q-function to evaluate the multi-time-step policy. This is complicated by the fact

that the temporally extended actions have much higher dimensionality, and in fact can have variable dimensionality if the "chunk" size is not held constant.

To resolve this issue, our first main idea in this work is to use generalized implicit Q-learning (IQL) (Kostrikov et al., 2022) backups with a V-function, which can handle such large and variable-size action spaces seamlessly. Instead of training an action-dependent Q-function and computing $\arg\max$ actions as in conventional value-based RL methods, we generalize IQL to directly learn a V-function with implicit Bellman updates via an asymmetric expectile loss, which approximates the $\max$ operator in the Bellman equation with no need for enumerating the high-dimensional temporally extended action space. Moreover, this generalization allows the agent to learn a value function from arbitrary-length action sequences, as this V-function backup only depends on states, which enables learning from tuples consisting of different action lengths. Our second idea is to use an advantage-weighted sequence modeling objective to extract a highly expressive Transformer policy from the value function. Specifically, we formulate the problem of policy learning as a *weighted* sequence modeling problem, and propose a way to extract a Transformer policy that produces temporally extended actions from the learned value function.

Through our experiments on six robotic manipulation benchmark environments, we show that our method is capable of learning a performant policy from suboptimal demonstrations, demonstrating the effectiveness of temporally extended actions and action chunking in RL. Furthermore, we show that our generalized IQL and advantage-weighted sequence modeling objectives, which does not commit to a fixed action length, can learn a policy even from time-heterogeneous datasets with multiple action frequencies, outperforming recently proposed techniques for handling variable frequency robotic data.

## 2 RELATED WORK

**Offline RL.** The goal of offline RL is to train a policy solely from a static dataset, without environment interactions (Lange et al., 2012; Levine et al., 2020). Previous work in offline RL has introduced various algorithms based on behavioral constrained optimization (Wu et al., 2019; Fujimoto et al., 2019; Fujimoto & Gu, 2021), conservative optimization (Kumar et al., 2020; Cheng et al., 2022), in-sample value maximization (Kostrikov et al., 2022; Xu et al., 2023; Garg et al., 2023), and generative models (Chen et al., 2021; Janner et al., 2021; 2022). In this work, we generalize a previous in-sample value maximization method, implicit Q-learning (Kostrikov et al., 2022), to make it compatible with both *temporally extended actions* and time-heterogeneous datasets. Related to our work, Burns et al. (2022) recently proposed an offline RL method based on conservative Q-learning (Kumar et al., 2020) that can deal with time-heterogeneous datasets by adaptively adjusting discount factors and $n$-step returns. While our work also generalizes offline value function learning to multiple time scales, our method can both learn from and produce *action chunks*, unlike Burns et al. (2022), which only considers single-step actions. We empirically show that the use of action chunks is crucial for performance in our benchmark environments, especially when the offline datasets consist of non-Markovian human demonstrations.

**Hierarchical RL.** The idea of using temporally extended actions has long been explored in the context of hierarchical RL (Sutton et al., 1999; Stolle & Precup, 2002; Bacon et al., 2017; Vezhnevets et al., 2017; Nachum et al., 2018) or action repetition (Lakshminarayanan et al., 2017; Sharma et al., 2017; Metelli et al., 2020; Park et al., 2021; Biedenkapp et al., 2021), whose goal is to reduce the number of decision steps via temporally extended high-level actions. Among hierarchical approaches, our method is conceptually related to offline hierarchical RL methods, such as offline skill extraction (Krishnan et al., 2017; Pertsch et al., 2020; Ajay et al., 2021; Shi et al., 2022; Jiang et al., 2023; Rosete-Beas et al., 2022) and hierarchical imitation learning (Lynch et al., 2019; Gupta et al., 2019), in that we also extract a sequence of actions from an offline dataset. However, unlike offline skill extraction methods, we do not have separate latent action spaces, and unlike hierarchical imitation learning methods, we do not require expert trajectories. Instead, we directly model action sequences using a Transformer with an advantage-weighted sequence modeling objective, which allows us to learn an action chunk policy even from suboptimal datasets, such as datasets consisting of a mixture of demonstrations and randomized trajectories.

**Sequential decision making with expressive policies.** Several works have been proposed that use expressive policy classes beyond unimodal Gaussian distributions. To enhance the expressivity of policies, previous works proposed to use Gaussian mixture models (Mandlekar et al., 2021), implicit functions (Florence et al., 2021), action discretization (Metz et al., 2017; Luo et al., 2023), diffusion models (Hansen-Estruch et al., 2023; Chi et al., 2023), and Transformers (Shafiullah et al., 2022; Zhao et al., 2023). In particular, several recent works (Shafiullah et al., 2022; Zhao et al., 2023; Chi et al., 2023) have found *action chunking* to be often beneficial for behavioral cloning. Instead of a single-step Markovian policy $\pi(a_t|s_t)$, they employ a sequential policy $\pi(a_{t:t+n-1}|s_t)$ that produces action sequences of length $n$. Our method is particularly related to these action chunking methods as we also train a sequential policy to model temporally extended actions. However, unlike these *behavioral cloning* approaches, our focus is on offline RL, which introduces the challenges of learning a value function and modulating (potentially suboptimal) dataset policies toward optimal behaviors.

## 3 PRELIMINARIES

**Problem setting.** We consider a Markov decision process (MDP) defined as $\mathcal{M} = (\mathcal{S}, \mathcal{A}, r, p, \gamma)$, where $\mathcal{S}$ is the state space, $\mathcal{A}$ is the action space, $r : \mathcal{S} \times \mathcal{A} \rightarrow \mathbb{R}$ is the reward function, $p : \mathcal{S} \times \mathcal{A} \rightarrow \mathcal{P}(\mathcal{S})$ is the transition dynamics function, and $\gamma$ is the discount factor. We assume that the given MDP $\mathcal{M}$ is derived from an original continuous-time MDP with a discretization time scale $\delta$ (*i.e.*, the control frequency is $1/\delta$). We refer to Doya (2000); Tallec et al. (2019) for a formal treatment of time-discretized MDPs. Our goal in this work is to learn a performant task policy $\pi(a|s)$ that maximizes the given reward function solely from an *offline* dataset $\mathcal{D}$. For some of our experiments, we consider a *time-heterogeneous* offline dataset that consists of transitions from diverse $\delta$-discretized MDPs with different $\delta$s.

**Implicit Q-learning (IQL).** Standard Q-learning algorithms train a Q function $Q(s, a)$ by minimizing the Bellman error, $(Q(s, a) - (r(s, a) + \gamma \max_{a' \in \mathcal{A}} Q(s', a')))^2$. However, in the *offline* RL setting, the $\arg \max$ action $a'$ in the Bellman equation may lie outside of the dataset support due to the insufficient coverage of the dataset. Since we cannot collect additional data or provide corrective feedback in the offline setting, the agent may exploit such out-of-distribution actions, which often leads to catastrophic overestimation of $Q$ values (Levine et al., 2020). To address this issue, Kostrikov et al. (2022) proposed implicit Q-learning (IQL), which replaces the $\max$ operator in the Bellman equation with an asymmetric expectile loss to approximate the maximum only using in-sample actions. Specifically, IQL trains a value function $V_\psi$ parameterized by $\psi$ and a Q function $Q_\theta$ parameterized by $\theta$ with the following loss functions:

$$\mathcal{L}_V(\psi) = \mathbb{E}_{(s,a)\sim\mathcal{D}}[L_2^\tau(V_\psi(s) - Q_{\bar{\theta}}(s, a))], \tag{1}$$

$$\mathcal{L}_Q(\theta) = \mathbb{E}_{(s,a,r,s')\sim\mathcal{D}}[(Q_\theta(s, a) - r - \gamma V_\psi(s'))^2], \tag{2}$$

where $L_2^\tau$ denotes the expectile loss with a parameter $\tau \in [0.5, 1)$, $L_2^\tau(x) = |\tau - \mathbb{1}(x < 0)|x^2$, and $\bar{\theta}$ denotes the target Q parameters (Mnih et al., 2013). After training the value function, IQL uses advantage-weighted regression (AWR) (Peng et al., 2019) to extract a policy from the value function. AWR maximizes the following objective:

$$J_\pi(\phi) = \mathbb{E}_{(s,a)\sim\mathcal{D}}[\exp(\beta \cdot (Q_\theta(s, a) - V_\psi(s))) \log \pi_\phi(a|s)], \tag{3}$$

where $\phi$ denotes the parameters of the policy $\pi_\phi(a|s)$ and $\beta$ denotes the temperature hyperparameter. Intuitively, Equation (3) extracts a policy via weighted behavioral cloning, where the weights are defined by the exponentiated advantages, hence biasing the policy toward actions that lead to high Q values.

**Implicit V-learning (IVL).** Several previous methods (Xu et al., 2022; Ghosh et al., 2023; Park et al., 2023) have used a value-only variant of IQL, which does not require fitting a Q function. We call this variant implicit V-learning (IVL). IVL maximizes the following objective:

$$\mathcal{L}_V(\psi) = \mathbb{E}_{(s,r,s')\sim\mathcal{D}}[L_2^\tau(V_\psi(s) - r - \gamma V_{\bar{\psi}}(s'))], \tag{4}$$

where $\bar{\psi}$ denotes the parameters of the target value network. Unlike Equation (1), Equation (4) directly takes the Bellman backup from $r + \gamma V_{\bar{\psi}}(s')$. We note that IVL may be optimistically biased

in stochastic environments (Kostrikov et al., 2022), as it takes the (implicit) maximum over not only actions but also the next states. Despite the limitation, IVL provides a simple, convenient way to learn a value function without the need for action-dependent Q-functions. Hence, we choose to employ IVL for our method for simplicity, following previous work (Xu et al., 2022; Ghosh et al., 2023; Park et al., 2023). We leave mitigating this optimism bias (*e.g.*, with Yang et al. (2023); Villaflor et al. (2022)) for future work.

## 4    V-FORMER

Our main goal in this paper is to propose an offline RL algorithm, which we call Value-Transformer (**V-Former**), that uses action sequence policies for handling challenging data distributions with suboptimal, multi-modal, and non-Markovian behavioral policies. V-Former consists of two main components. In Section 4.1, we generalize implicit V-learning to enable learning from arbitrary action sequences. In Section 4.2, we describe a way to extract a Transformer policy from the learned value function.

### 4.1    VALUE LEARNING FROM ARBITRARY-LENGTH ACTIONS

We will first discuss how V-Former enables value function learning with action sequences of arbitrary length or frequency, corresponding to variable discretization time scales. One naïve way to deal with length-$n$ actions is to extend the action space $\mathcal{A}$ to $\mathcal{A}^n$ and then run a standard offline RL algorithm. However, this not only requires a Q-function to predict values for multi-step actions, where the input space grows exponentially as the action length increases, but also may necessitate enumerating over the exponentially large number of actions to compute the maximum in the Bellman operator. To address this challenge, our first main idea in this work is to generalize an in-sample value maximization algorithm, implicit V-learning (IVL, Section 3), which approximately computes the maximum in the Bellman operator by minimizing an asymmetric expectile loss, without the need to explicitly maximize over the action space. To generalize IVL to arbitrary-length transitions, we begin with the following recursive expansion of the Bellman equation:

$$V(s_t) = \max_{a_t, s_{t+1}} r_t + \gamma V(s_{t+1}) \tag{5}$$

$$= \max_{\substack{a_{t:t+1}, \\ s_{t+1:t+2}}} r_t + \gamma r_{t+1} + \gamma^2 V(s_{t+2}) \tag{6}$$

$$= \cdots \tag{7}$$

$$= \max_{\substack{a_{t:t+n-1}, \\ s_{t+1:t+n}}} \sum_{i=0}^{n-1} \gamma^i r_{t+i} + \gamma^n V(s_{t+n}), \tag{8}$$

where the $\max$ is taken over the feasible state-action tuples that are reachable from $s_t$. Now, similarly to IVL (Equation (1)), we replace the $\max$ operator in Equation (8) with the expectile loss with a learnable value function $V_\psi(s)$ parameterized by $\psi$:

$$\mathcal{L}_V(\psi) = \mathbb{E}_{(s_{t:t+n}, r_{t:t+n-1}) \sim \mathcal{D}} \left[ L_2^\tau \left( V_\psi(s_t) - \sum_{i=0}^{n-1} \gamma^i r_{t+i} - \gamma^n V_{\bar{\psi}}(s_{t+n}) \right) \right]. \tag{9}$$

Intuitively, Equation (9) aims to regress $V_\psi(s_t)$ toward the "$n$-step return" of $\sum_{i=0}^{n-1} \gamma^i r_{t+i} + \gamma^n V_{\bar{\psi}}(s_{t+n})$ only when the target value is greater than the current value estimate. We note that, unlike the standard $n$-step updates based on the square loss (*e.g.*, Burns et al. (2022)), which could lead to suboptimal values when the $n$-step return is not optimal, Equation (9) leads $V_\psi(s)$ to the optimal value function $V^*(s)$ under the dataset support constraints in deterministic environments (when $\tau \to 1$), thanks to the expectile regression which approximates the $\max$ operator.

Finally, we generalize Equation (9) to handle multiple discretization time-scales $\delta$ as follows:

$$\mathcal{L}_V(\psi) = \mathbb{E}_{(\delta, s_{t:t+n}, r_{t:t+n-1}) \sim \mathcal{D}} \left[ L_2^\tau \left( V_\psi(s_t) - \sum_{i=0}^{n-1} \bar{\gamma}^{i\delta} r_{t+i} - \bar{\gamma}^{n\delta} V_{\bar{\psi}}(s_{t+n}) \right) \right], \tag{10}$$

where $(\delta, s_{t:t+n}, r_{t:t+n-1})$ denotes the trajectory chunk $(s_t, r_t, s_{t+1}, \ldots, r_{t+n-1}, s_{t+n})$ with a discretization time scale of $\delta$. Note that we adjust the discount factor as $\bar{\gamma}^\delta$ to correctly handle different

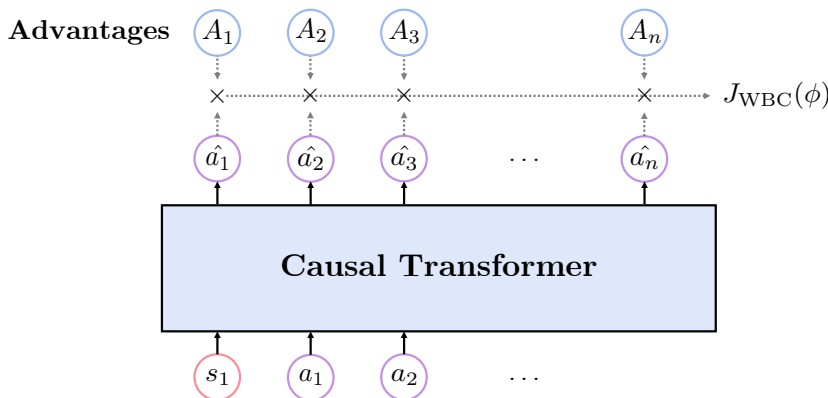

Figure 1: **Illustration of the weighted sequence modeling objective of V-Former.**

$\delta$s, where $\bar{\gamma}$ is a continuous-time discount factor (Tallec et al., 2019). Due to the in-sample maximization property of IVL, $V_\psi(s)$ trained by Equation (10) still approximates $V^*(s)$ under the dataset support constraints. As a result, we can train a value function from arbitrary action frequencies or action lengths with Equation (10), which allows us to leverage multiple datasets with different discretization time scales.

## 4.2 ADVANTAGE-WEIGHTED REGRESSION WITH A TRANSFORMER POLICY

Our next step is to extract a sequential Transformer policy that produces action chunks from the learned value function $V_\psi(s)$. We first describe the vanilla Transformer objective for naïvely cloning dataset actions of length $n$:

$$J_{\text{BC}}(\phi) = \mathbb{E}_{(s_t, a_{t:t+n-1}) \sim \mathcal{D}} \left[ \log \pi_\phi(a_t, \ldots, a_{t+n-1} | s_t) \right] \tag{11}$$

$$= \mathbb{E}_{(s_t, a_{t:t+n-1}) \sim \mathcal{D}} \left[ \sum_{i=0}^{n-1} \log \pi_\phi(a_{t+i} | s_t, a_{t:t+i-1}) \right], \tag{12}$$

where we autoregressively decompose the log probability $\log \pi_\phi(a_t, \ldots, a_{t+n-1} | s_t)$ with $n$ terms. This is the same as the typical generative sequence modeling objective (Brown et al., 2020), except that it is conditioned on the state $s_t$. To bias the actions toward states that have high values, inspired by AWR (Peng et al., 2019) and CRR (Wang et al., 2020), we propose the following *advantage-weighted* objective for our Transformer policy:

$$J_{\text{WBC}}(\phi) = \mathbb{E}_{(s_{t:t+n}, a_{t:t+n-1}) \sim \mathcal{D}} \left[ \sum_{i=0}^{n-1} \log \pi_\phi(a_{t+i} | s_t, a_{t:t+i-1}) f(A_{t+i}) \right], \tag{13}$$

where we highlight the difference from Equation (12) in blue. The advantage $A_{t+i}$ is defined as $r_{t+i} + \gamma V_\psi(s_{t+i+1}) - V_\psi(s_{t+i})$, and $f$ is a non-negative, non-decreasing function. If $n = 1$ and $f(x) = \exp(x/\beta)$, Equation (13) becomes the regular AWR objective (Peng et al., 2019) with a temperature $\beta$, and if $n = 1$ and $f(x) = \mathbb{1}(x \geq \beta)$, it becomes the binary CRR objective (Wang et al., 2020) with a threshold $\beta$. Similarly to AWR and CRR, Equation (13) biases the action *sequence* distribution toward high-value states based on the advantage weights $f(A_{t+i})$ (Figure 1).

**Implementation.** V-Former has two trainable components: a value function $V_\psi(s)$ and a Transformer policy $\pi_\phi(a_{t:t+n-1} | s_t)$. We train the value function with length-$n$ sized trajectory chunks, where $n$ is randomly sampled between 1 and $N$, and the Transformer policy with length-$N$ sized action chunks (*i.e.*, the maximum length). Following the practice of previous Transformer-based RL methods (Janner et al., 2021; Shafiullah et al., 2022), we discretize the action space per dimension and minimize the cross entropy loss when training the Transformer policy. To wait until the value function converges, we start the training of the Transformer policy after $E$ gradient steps. We provide a pseudocode for V-Former in Algorithm 1 and the full implementation details in Appendix A.

---

**Algorithm 1** V-Former

---

1: **Input**: offline dataset $\mathcal{D}$
2: Initialize value function $V_\psi(s)$, Transformer policy $\pi_\phi(a_{t:t+n-1}|s_t)$, learning rates $\lambda_V, \lambda_\pi$
3: **while** not converged **do**
4:     Sample $n$ from $\mathrm{Unif}(\{1, 2, \ldots, N\})$
5:     $\psi \leftarrow \psi - \lambda_V \nabla_\psi \mathcal{L}_V(\psi)$ with $(\delta, s_{t:t+n}, r_{t:t+n-1}) \sim \mathcal{D}$
6:     **if** (# total gradient steps) $> E$ **then**
7:         $\phi \leftarrow \phi + \lambda_\pi \nabla_\phi J_{\mathrm{WBC}}(\phi)$ with $(s_{t:t+N}, a_{t:t+N-1}) \sim \mathcal{D}$
8:     **end if**
9: **end while**

---

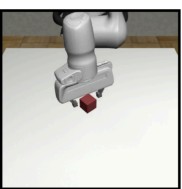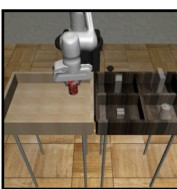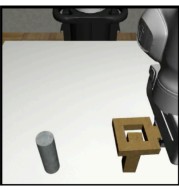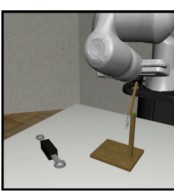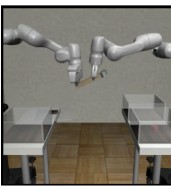

Figure 2: **Five Robomimic tasks: `lift`, `can`, `square`, `tool-hang`, and `transport`.**

**Evaluation.** At evaluation time, we consider length-$k$ open-loop control, *i.e.*, we execute only the first $k$ actions of the action sequence, and query the policy every $k$ steps. This execution scheme becomes equivalent to closed-loop control when $k = 1$ and fully open-loop control when $k = N$. In general, we expect that open-loop control is better than closed-loop control when the behavioral policies are highly non-Markovian, since fully closed-loop actions at evaluation time cannot simulate a non-Markovian policy. This is also observed by Zhao et al. (2023). Otherwise, we expect closed-loop control to be better than open-loop control because it can be more reactive to observations.

## 5 EXPERIMENTS

In our experiments, we aim to answer the following questions:

- Does V-Former lead to better performance on suboptimal, non-Markovian, and multi-modal datasets compared to the baselines without action chunking or advantage weighting?
- Can V-Former learn a value function and a performant task policy from time-heterogeneous datasets?

### 5.1 RESULTS ON ROBOTIC MANIPULATION TASKS

**Baselines.** We mainly compare V-Former ("VF") with two baselines obtained by ablating our components: (1) V-Former without action chunks (Equation (13) with $n = 1$) and (2) Transformer without advantage weights ("BC", Equation (12)). To ensure a fair comparison, we implement our method and the baselines on the same codebase, and apply the same Transformer architecture and action discretization scheme for all the baselines.

**Environments and datasets.** For our main quantitative evaluation, we employ the five tasks from the Robomimic benchmark (Mandlekar et al., 2021): `lift`, `can`, `square`, `tool-hang`, and `transport` (Figure 2). The goal of these tasks is to manipulate a 7-DoF robot arm (or two robot arms for `transport`) to achieve the desired outcomes: *e.g.*, in `transport`, the agent must learn bimanual maneuvers to transfer a hammer from a closed container on a shelf to the target bin on another shelf; in `tool-hang`, the robot arm must learn high-precision manipulation behaviors to assemble a frame by inserting a hook into a narrow base. For datasets, we employ human demonstrations ("Proficient-Human (PH)"), which feature narrow, multi-modal, and non-Markovian behaviors. It has been previously reported that such human-generated datasets cause offline RL methods to struggle (Mandlekar et al., 2021). We additionally constructed suboptimal versions of these

Table 1: **Average success rate on the five Robomimic tasks (*expert* datasets).** VF denotes V-Former and BC denotes V-Former without advantage weights.

| Method $(N, k)$ | BC $(1, 1)$ | BC $(3, 1)$ | BC $(3, 3)$ | VF $(1, 1)$ | VF $(3, 1)$ | VF $(3, 3)$ **(ours)** |
|---|---|---|---|---|---|---|
| lift | 88.7 | 93.3 | **94.7** | **94.7** | 94 | 93.3 |
| can | 90 | 88.7 | 91.3 | 92.7 | 90.7 | **95.3** |
| square | 62.7 | **79.3** | 78 | 65.3 | 73.3 | 67.3 |
| transport | 22.7 | 13.3 | **35.3** | 23.3 | 20.7 | 32 |
| tool-hang | 14.7 | 22 | 36.7 | 14.7 | 19.3 | **38** |
| Average | 56.7 | 59.3 | **67.2** | 58.1 | 59.6 | 65.2 |

Table 2: **Average success rate on the five Robomimic tasks (*suboptimal* datasets).** VF denotes V-Former and BC denotes V-Former without advantage weights.

| Method $(N, k)$ | BC $(1, 1)$ | BC $(3, 1)$ | BC $(3, 3)$ | VF $(1, 1)$ | VF $(3, 1)$ | VF $(3, 3)$ **(ours)** |
|---|---|---|---|---|---|---|
| lift | 50 | 60.7 | 58.7 | 58.7 | 61.3 | **70** |
| can | 52 | 58 | 59.3 | 54.7 | **70** | 66.7 |
| square | 37.3 | 50.7 | **66** | 47.3 | 56 | **66** |
| transport | 8 | 10.7 | 16 | 14.7 | 12.7 | **25.3** |
| tool-hang | 11.3 | 18 | 28 | 15.3 | 21.3 | **28.7** |
| Average | 31.7 | 39.6 | 45.6 | 38.1 | 44.3 | **51.3** |

datasets, where we might expect both more opportunities for improvement (due to better coverage), but also potentially greater challenges for offline RL methods (due to the mixture of two different data distributions). Specifically, for each task, we collect a suboptimal dataset by combining the 200 trajectories in the corresponding PH dataset with 200 additional randomized trajectories. In these environments, we use an action chunk size of 3 (*i.e.*, $N = 3$) for V-Former and repeat each experiment with three random seeds.

**Results.** Table 1 demonstrates the full results on the five Proficient-Human (PH) datasets. The results suggest that action chunking with open-loop control is generally better than both action chunking with closed-loop control and vanilla single-step policies across the five tasks. These results coincide with the previous observation by Mandlekar et al. (2021) that vanilla BC policies led to worse performance than the recurrent counterparts. This is likely because the dataset distributions are narrow and the behavioral policies (*i.e.*, human demonstrations) are non-Markovian. On these

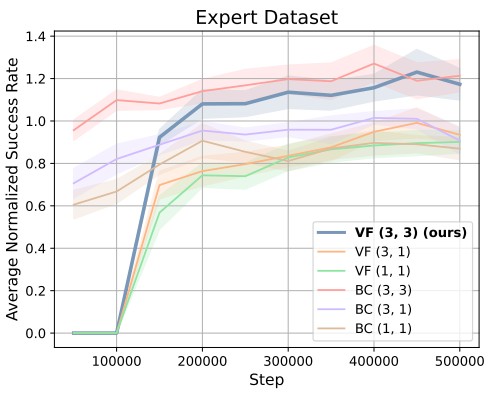
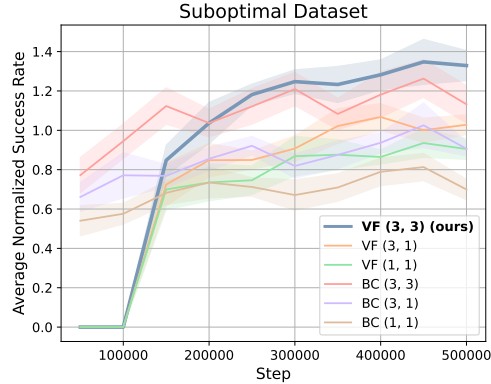

Figure 3: **Aggregated normalized success rates on the two types of Robomimic datasets.** Each line is averaged over five tasks and three seeds (*i.e.*, 15 runs in total).

Table 3: **Average returns on Kitchen (time-heterogeneous dataset).** VF denotes V-Former and BC denotes V-Former without advantage weights.

| Task | Naïve Mixing | Adaptive $N$-Step | VF ($N = 12, k = 1$) **(ours)** |
|---|---|---|---|
| Kitchen ($\delta = 40$) | 20.2 | 34.6 | **39.6** |
| Kitchen ($\delta = 30$) | 9.3 | 19.9 | **42.1** |

PH datasets, however, RL agents do not necessarily exhibit better performances than BC agents, since the datasets consist of expert trajectories.

We then evaluate V-Former on the more challenging suboptimal datasets consisting of a mixture of demonstrations and randomized trajectories. Table 2 demonstrates the full results, where V-Former outperforms all of the baselines in most of the tasks. These results suggest that our method is capable of learning a performant task policy even when the dataset is suboptimal, thanks to our policy extraction scheme based on the learned value function. Moreover, by comparing the open-loop and closed-loop results (*i.e.*, VF $(3, 3)$ vs. VF $(3, 1)$), we confirm that action chunking with open-loop control is beneficial for RL as well. Figure 3 shows the aggregated training curves of V-Former and five baselines on both types of datasets. These plots show the average *normalized* success rate for each of the six methods. To account for differences in difficulties between the five tasks, the success rate per task is divided by the average task performance (*across* the six methods) before aggregation. We note that "VF $(3, 3)$", *i.e.*, our V-Former, achieves the best or near-best performance in both settings.

## 5.2 RESULTS ON TIME-HETEROGENEOUS DATASETS

Next, to verify whether V-Former can learn a value function from time-heterogeneous datasets with multiple action frequencies, we evaluate our method on the Franka Kitchen environment by Gupta et al. (2019) using the time-heterogeneous datasets used by Burns et al. (2022). Franka Kitchen is an environment in which a 9-DoF Franka robot is placed in a kitchen with multiple objects, where the goal is to complete four subtasks: open the microwave, move the kettle, flip the light switch, and open the slide cabinet door. The time-heterogenous dataset from Burns et al. (2022) contains a mixture of data of two different action frequencies: $\delta = 30$ and $\delta = 40$. The data with $\delta = 40$, the default for Kitchen, comes directly from

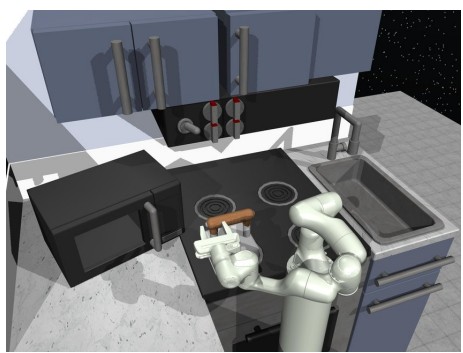

Figure 4: **The Franka Kitchen environment.**

the `kitchen-complete-v0` D4RL dataset, consisting of demonstrations of all four target subtasks being completed in order. The data with $\delta = 30$ comes from rolling out a trained policy on the environment with the modified action frequency and consists of 100000 transitions. We train V-Former with $N = 12$ on the time-heterogeneous dataset and evaluate it on both the environments with $\delta = 30$ and $\delta = 40$. For comparisons, we consider the adaptive $N$-step method proposed by Burns et al. (2022), which can correctly handle time-heterogeneous datasets by adjusting discount factors and rewards, as well as the "naïve mixing" baseline used by Burns et al. (2022), which simply runs an offline RL algorithm without adjusting the difference in discretization time scales. Table 3 demonstrates the comparison results, where we take the baseline performances from Burns et al. (2022). The results suggest that V-Former is capable of learning a value function and policy from time-heterogeneous datasets with multiple frequencies, outperforming the previous baselines in both settings.

## 5.3 ABLATION STUDY

**Action sequence lengths.** To confirm the effect of the maximum action length $N$ on performance, we conduct an ablation study of different action chunk sizes and different open-loop action lengths at

Table 4: **Evaluating V-Former on the Robomimic Proficient-Human (PH) `tool-hang` dataset.** VF denotes V-Former and BC denotes V-Former without advantage weights.

| Method $(N, k)$ | VF $(1, 1)$ | VF $(3, 1)$ | VF $(3, 3)$ | VF $(5, 1)$ | VF $(5, 3)$ | VF $(8, 1)$ | VF $(8, 3)$ |
|---|---|---|---|---|---|---|---|
| `tool-hang` | 14.7 | 19.3 | **38** | 14.7 | 37.3 | 6.7 | 22 |

evaluation time. We consider action chunk sizes $(N)$ of $1$, $3$, $5$, and $8$, and open-loop action lengths $(k)$ of $1$ and $3$. Table 4 shows the results on the `tool-hang` task from the Robomimic benchmark, where we find $(N, k) = (3, 3)$ to perform the best. Also, the results suggest that open-loop control always leads to better performance than closed-loop control in this task, regardless of the maximum action chunk size $N$.

## 6 CONCLUSION

In this work, we proposed V-Former, an offline RL method that can both learn from and produce temporally extended actions. Specifically, we generalized implicit Q-learning backups to enable value functions to be learned from trajectory chunks of arbitrary action lengths, and proposed a weighted sequence modeling objective to extract a Transformer policy from the learned value function. Throughout our evaluation on five robotic manipulation tasks with challenging data distributions consisting of suboptimal, multi-modal, and non-Markovian behavioral policies, we demonstrated that V-Former exhibits strong performance by utilizing temporally extended actions. Furthermore, we showed that V-Former is capable of learning a value function from time-heterogeneous datasets with multiple frequencies, outperforming previous methods.

### REPRODUCIBILITY STATEMENT

We will release our implementation for V-Former upon acceptance. We provide the full implementation details in Appendix A.

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

## A  IMPLEMENTATION DETAILS

We provide the full list of hyperparameters we use for V-Former in Tables 5 and 6. For the Robomimic tasks, we subtract 1 from the original rewards as we found this to perform better than the original 0/1 reward scheme, similarly to Kostrikov et al. (2022). We use the Binary CRR objective with an advantage threshold of $-0.5$ and use $E = 100K$. We evaluate performance with 50 episodes every 50K steps, with the evaluation numbers at 500K steps used for the results. For experiments with time-heterogeneous datasets, we use the AWR objective with a temperature of 1, with $E = 0$.

Table 5: V-Former hyperparameters for Robomimic experiments.

| Hyperparameter | Value |
| --- | --- |
| Learning rate | 0.0003 |
| Learning rate decay | Cosine (actor only) |
| Optimizer | Adam (Kingma & Ba, 2015) |
| Normalization | LayerNorm (Ba et al., 2016) |
| Batch size | 256 |
| Training steps | 500000 |
| Value pretraining steps | 100000 |
| Discount | 0.99 |
| Expectile | 0.7 |
| Target network smoothing coefficient | 0.005 |
| Action discretization bins | 256 |
| Transformer layers | 4 |
| Transformer heads | 4 |
| Transformer MLP dimension | 512 |
| Transformer embedding dimension | 512 |
| Transformer dropout rate | 0.1 |
| Value network dimensions | (256, 256) |
| Binary CRR advantage threshold | $-0.5$ |

Table 6: V-Former hyperparameters for time-heterogeneous Kitchen experiments.

| Hyperparameter | Value |
| --- | --- |
| Learning rate | 0.0003 |
| Learning rate decay | Cosine (actor only) |
| Optimizer | Adam (Kingma & Ba, 2015) |
| Normalization | LayerNorm (Ba et al., 2016) |
| Batch size | 256 |
| Training steps | 500000 |
| Discount | 0.99 |
| Expectile | 0.7 |
| Temperature | 1 |
| Target network smoothing coefficient | 0.005 |
| Action discretization bins | 256 |
| Transformer layers | 2 |
| Transformer heads | 2 |
| Transformer MLP dimension | 256 |
| Transformer embedding dimension | 256 |
| Transformer dropout rate | 0.1 |
| Value network dimensions | (256, 256) |

