# OpenReview forum: "V-Former: Offline RL with Temporally-Extended Actions"
_ICLR.cc/2024/Conference — Submitted to ICLR 2024_

### Official Review · Reviewer_ZdXy · 2023-10-26

**Soundness:** 2 fair
**Presentation:** 3 good
**Contribution:** 2 fair
**Rating:** 5
**Confidence:** 4

**Summary:**

This paper proposes a transformer-based offline RL method. It first introduces an "implicit V-learning" algorithm (similar to IQL) that can be extended to multiple timesteps. It then learns a transformer-based policy via a weighted behavior cloning objective, where the weight depends on the temporally extended learned value function. The method is evaluated on several continuous control benchmarks including the Robomimic and Franka Kitchen.

**Strengths:**

**originality**: Although the idea of "implicit V-learning" has been used in several prior works, this paper proposes to extend the learning objective to multiple timesteps. Moreover, the combination of IVL and transformer-based policy learning is novel.

**clarity**: I find this paper generally well written and easy to follow.

**Weaknesses:**

My main concern is on the experiments, which I don't find sufficient enough to demonstrate the strength of the proposed method as an offline RL algorithm.

First, the evaluation was conducted against some variants of the proposed method (which seems more like ablation studies to me) but didn't consider any existing offline RL baselines (which I don't see any limitations in the settings that prevent one from doing so).

Second, the experiments were conducted only on expert datasets, and *suboptimal datasets* (expert + random data) which were a bit artificial. While it's known that the performance of offline RL / imitation learning methods varies drastically depending on the data quality, it's important to evaluate the method on datasets of various optimality, and especially on those which are similar to real-world settings, e.g. the multi-human datasets from Robomimic.

Lastly, the proposed method seems sensitive (task depedent) to certain critical hyperparameters, including the "action chunk size". But experiments only cover a small range of those parameters. I believe more extensive ablation studies would be helpful to show if the method is robust and generally applicable to various continuous control problems.

**Questions:**

1. Why not including established baselines, e.g. BC-RNN, BCQ, CQL (which were used as baselines in the Robomimic paper), and transformer-based baselines like Decision Transformer and Trajectory Transformer?

2. How are the success rates in Fig 3 normalized? Why are some greater than 1?

3. The expert + random datasets seem a bit artificial. Why not evaluating on the existing multi-human datasets (which were generated by human operators of different level of proficiency on the tasks) from Robomimic instead?

4. What's your intuition on selecting an optimal range of N?

5. As noted in the appendix, different weight functions f(x) were used in the Robomimic and FrankaKitchen experiments. It would be nice to include an ablation table for both f(x) on both environments to show how sensitive the method is to f(x).

6. Have you tried evaluating the method on tasks with discrete action spaces, e.g. maze?

---

> ### Author Response · Authors · 2023-11-19
>
> We thank the reviewer for the thorough review and constructive feedback about this work. Below, we describe how we have added an additional comparison with $3$ existing offline RL baselines, new results on different types of suboptimal datasets, and a new ablation study with different action chunk sizes. We believe that these changes strengthen the paper, and welcome additional suggestions for further improving the work.
>
> * **$\mathbf{3}$ new baselines - CQL, IQL, and DT**
>
> Thank you for the suggestion. We would like to first note that we have (already) included comparisons with three existing offline RL and BC methods, (1) BC, (2) BC+Transformer, and (3) IQL (which correspond to $BC (1, 1)$, $BC (3, 3)$, and $VF (1, 1)$, respectively (Tables 1, 2, Fig. 3); these methods use the same action discretization as V-Former). However, following the suggestion, below, we compare V-Former with $\mathbf{3}$ additional existing offline RL methods: **CQL**, **(Original) IQL**, and **Decision Transformer**.
>
> **(1) Results on expert (PH) datasets:**
> | Task | BC | BC+Transformer | IQL | CQL | (Original) IQL | Decision Transformer | V-Former (ours) |
> | :----: | :----: | :----: | :----: | :----: | :----: | :----: | :----: |
> | $\texttt{can}$ | $90.0$ $\tiny{\pm 3.5}$ | $91.3$ $\tiny{\pm 9.0}$ | $92.7$ $\tiny{\pm 6.1}$ | $10.0$ $\tiny{\pm 11.1}$ | $37.3$ $\tiny{\pm 9.5}$ | $93.3$ $\tiny{\pm 1.2}$ | $\mathbf{95.3}$ $\tiny{\pm 3.1}$ |
> | $\texttt{lift}$ | $88.7$ $\tiny{\pm 9.5}$ | $94.7$ $\tiny{\pm 2.3}$ | $94.7$ $\tiny{\pm 4.2}$ | $70.7$ $\tiny{\pm 11.5}$ | $74.0$ $\tiny{\pm 8.7}$ | $\mathbf{98.0}$ $\tiny{\pm 2.0}$ | $93.3$ $\tiny{\pm 3.1}$ |
> | $\texttt{square}$ | $62.7$ $\tiny{\pm 5.0}$ | $\mathbf{78.0}$ $\tiny{\pm 7.2}$ | $65.3$ $\tiny{\pm 8.1}$ | $0.0$ $\tiny{\pm 0.0}$ | $31.3$ $\tiny{\pm 9.0}$ | $30.7$ $\tiny{\pm 6.4}$ | $67.3$ $\tiny{\pm 1.2}$ |
> | $\texttt{tool\\_hang}$ | $14.7$ $\tiny{\pm 8.3}$ | $36.7$ $\tiny{\pm 8.1}$ | $14.7$ $\tiny{\pm 5.8}$ | $0.0$ $\tiny{\pm 0.0}$ | $6.0$ $\tiny{\pm 4.0}$ | $4.0$ $\tiny{\pm 2.0}$ | $\mathbf{38.0}$ $\tiny{\pm 7.2}$ |
> | $\texttt{transport}$ | $22.7$ $\tiny{\pm 3.1}$ | $\mathbf{35.3}$ $\tiny{\pm 8.1}$ | $23.3$ $\tiny{\pm 5.0}$ | $0.0$ $\tiny{\pm 0.0}$ | $2.0$ $\tiny{\pm 0.0}$ | $2.0$ $\tiny{\pm 2.0}$ | $32.0$ $\tiny{\pm 7.2}$ |
> | **average** | $55.7$ | $\mathbf{67.2}$ | $58.1$ | $16.1$ | $30.1$ | $45.6$ | $65.2$ |
>
> **(2) Results on suboptimal (mixed) datasets:**
> | Task | BC | BC+Transformer | IQL | CQL | (Original) IQL | Decision Transformer | V-Former (ours) |
> | :----: | :----: | :----: | :----: | :----: | :----: | :----: | :----: |
> | $\texttt{can}$ | $52.0$ $\tiny{\pm 7.2}$ | $59.3$ $\tiny{\pm 10.3}$ | $54.7$ $\tiny{\pm 10.3}$ | $0.0$ $\tiny{\pm 0.0}$ | $10.7$ $\tiny{\pm 8.3}$ | $33.3$ $\tiny{\pm 17.5}$ | $\mathbf{66.7}$ $\tiny{\pm 5.8}$ |
> | $\texttt{lift}$ | $50.0$ $\tiny{\pm 9.2}$ | $58.7$ $\tiny{\pm 4.2}$ | $58.7$ $\tiny{\pm 9.9}$ | $0.0$ $\tiny{\pm 0.0}$ | $26.0$ $\tiny{\pm 4.0}$ | $60.0$ $\tiny{\pm 3.5}$ | $\mathbf{70.0}$ $\tiny{\pm 15.6}$ |
> | $\texttt{square}$ | $37.3$ $\tiny{\pm 9.5}$ | $\mathbf{66.0}$ $\tiny{\pm 8.0}$ | $47.3$ $\tiny{\pm 5.8}$ | $0.0$ $\tiny{\pm 0.0}$ | $18.0$ $\tiny{\pm 6.0}$ | $6.0$ $\tiny{\pm 7.2}$ | $\mathbf{66.0}$ $\tiny{\pm 3.5}$ |
> | $\texttt{tool\\_hang}$ | $11.3$ $\tiny{\pm 5.0}$ | $28.0$ $\tiny{\pm 14.4}$ | $15.3$ $\tiny{\pm 4.2}$ | $0.0$ $\tiny{\pm 0.0}$ | $0.0$ $\tiny{\pm 0.0}$ | $0.0$ $\tiny{\pm 0.0}$ | $\mathbf{28.7}$ $\tiny{\pm 7.0}$ |
> | $\texttt{transport}$ | $8.0$ $\tiny{\pm 2.0}$ | $16.0$ $\tiny{\pm 6.9}$ | $14.7$ $\tiny{\pm 5.0}$ | $0.0$ $\tiny{\pm 0.0}$ | $0.0$ $\tiny{\pm 0.0}$ | $0.0$ $\tiny{\pm 0.0}$ | $\mathbf{25.3}$ $\tiny{\pm 2.3}$ |
> | **average** | $31.7$ | $45.6$ | $38.1$ | $0.0$ | $10.9$ | $19.9$ | $\mathbf{51.3}$ |
>
> The table above shows the comparison results of V-Former with three additional baselines on both expert and mixed (suboptimal) datasets (at 500K steps, 3 seeds each, $\pm$ denotes standard deviations). The results suggest that V-Former mostly outperforms the three new baselines, especially on challenging mixed datasets. In particular, previous offline RL methods that do not use temporally extended actions (i.e., CQL and IQL) struggle on these narrow, suboptimal datasets since their policies cannot fully represent highly non-Markovian and multi-modal behavioral policies. We will add these results to the paper.

---

> ### Author Response · Authors · 2023-11-19
>
> * **“The expert + random datasets seem a bit artificial.”**, **Evaluation on multi-human datasets**
>
> Thank you for the suggestion. Following the suggestion, we evaluate V-Former on **two** additional types of datasets: (1) the original multi-human (MH) datasets from Robomimic and (2) more “natural” suboptimal datasets consisting of original expert trajectories and additional “diversity” trajectories, where each diversity trajectory is obtained by concatenating initial steps from an expert trajectory to another goal-reaching trajectory toward a randomly sampled goal.
>
> **(1) Results on multi-human (MH) datasets**
>
> | Task | BC | BC+Transformer | IQL | V-Former (ours) |
> | :----: | :----: | :----: | :----: | :----: |
> | $\texttt{can}$ | $\mathbf{82.7}$ $\tiny{\pm 4.2}$ | $81.3$ $\tiny{\pm 4.6}$ | $80.7$ $\tiny{\pm 3.1}$ | $80.0$ $\tiny{\pm 5.3}$ |
> | $\texttt{lift}$ | $92.7$ $\tiny{\pm 1.2}$ | $94.7$ $\tiny{\pm 3.1}$ | $94.7$ $\tiny{\pm 3.1}$ | $\mathbf{97.3}$ $\tiny{\pm 3.1}$ |
> | $\texttt{square}$ | $32.7$ $\tiny{\pm 4.6}$ | $\mathbf{40.7}$ $\tiny{\pm 1.2}$ | $33.3$ $\tiny{\pm 9.5}$ | $\mathbf{40.7}$ $\tiny{\pm 5.0}$ |
> | $\texttt{transport}$ | $2.7$ $\tiny{\pm 3.1}$ | $4.7$ $\tiny{\pm 1.2}$ | $2.0$ $\tiny{\pm 2.0}$ | $\mathbf{8.7}$ $\tiny{\pm 3.1}$ |
> | **average** | $52.7$ | $55.3$ | $52.7$ | $\mathbf{56.7}$ |
>
> **(2) Results on more "natural" suboptimal datasets**
>
> | Task | BC | BC+Transformer | IQL | V-Former (ours) |
> | :----: | :----: | :----: | :----: | :----: |
> | $\texttt{can}$ | $7.3$ $\tiny{\pm 1.2}$ | $10.0$ $\tiny{\pm 4.0}$ | $6.7$ $\tiny{\pm 6.4}$ | $\mathbf{13.3}$ $\tiny{\pm 8.1}$ |
> | $\texttt{lift}$ | $10.7$ $\tiny{\pm 4.2}$ | $20.0$ $\tiny{\pm 2.0}$ | $\mathbf{40.0}$ $\tiny{\pm 2.0}$ | $38.7$ $\tiny{\pm 7.0}$ |
> | $\texttt{square}$ | $4.7$ $\tiny{\pm 1.2}$ | $4.0$ $\tiny{\pm 2.0}$ | $12.0$ $\tiny{\pm 5.3}$ | $\mathbf{13.3}$ $\tiny{\pm 5.0}$ |
> | $\texttt{tool\\_hang}$ | $0.7$ $\tiny{\pm 1.2}$ | $6.0$ $\tiny{\pm 4.0}$ | $2.7$ $\tiny{\pm 1.2}$ | $\mathbf{6.7}$ $\tiny{\pm 5.0}$ |
> | $\texttt{transport}$ | $5.3$ $\tiny{\pm 3.1}$ | $6.0$ $\tiny{\pm 5.3}$ | $7.3$ $\tiny{\pm 5.0}$ | $\mathbf{7.3}$ $\tiny{\pm 1.2}$ |
> | **average** | $5.7$ | $9.2$ | $13.7$ | $\mathbf{15.9}$ |
>
> The tables above compare V-Former with three baselines on the two additional datasets (at 500K steps, 3 seeds each, $\pm$ denotes standard deviations). The results suggest that, in both the MH datasets and the new suboptimal datasets, V-Former achieves the best or near-best performance, showing the effectiveness of temporally extended actions in offline RL. We will add these results to the final version of the paper.

---

> > ### Author Response · Authors · 2023-11-19
> >
> > * **How to select the action chunk size $N$?**
> >
> > As the reviewer pointed out, the action chunk size $N$ is a hyperparameter that we need to tune, as in most previous works in hierarchical RL and multi-step BC. However, we found that the optimal action chunk size $N$ is *not* very sensitive to individual tasks. We present the ablation results of V-Former (VF) on the five environments from Robomimic below:
> >
> > | Method ($N$, $k$) | VF (1, 1) | VF (3, 1) | VF (3, 3) | VF (5, 1) | VF (5, 3) | VF (8, 1) | VF (8, 3) |
> > | :----: | :----: | :----: | :----: | :----: | :----: | :----: | :----: |
> > | $\texttt{can}$ | $92.7$ $\tiny{\pm 6.1}$ | $90.7$ $\tiny{\pm 3.1}$ | $\mathbf{95.3}$ $\tiny{\pm 3.1}$ | $91.3$ $\tiny{\pm 4.2}$ | $91.3$ $\tiny{\pm 2.3}$ | $92.0$ $\tiny{\pm 3.5}$ | $94.7$ $\tiny{\pm 2.3}$ |
> > | $\texttt{lift}$ | $\mathbf{94.7}$ $\tiny{\pm 4.2}$ | $94.0$ $\tiny{\pm 2.0}$ | $93.3$ $\tiny{\pm 3.1}$ | $93.3$ $\tiny{\pm 3.1}$ | $\mathbf{94.7}$ $\tiny{\pm 1.2}$ | $43.3$ $\tiny{\pm 8.3}$ | $71.3$ $\tiny{\pm 8.1}$ |
> > | $\texttt{square}$ | $65.3$ $\tiny{\pm 8.1}$ | $73.3$ $\tiny{\pm 3.1}$ | $67.3$ $\tiny{\pm 1.2}$ | $66.0$ $\tiny{\pm 8.7}$ | $\mathbf{75.3}$ $\tiny{\pm 6.1}$ | $68.0$ $\tiny{\pm 13.1}$ | $64.7$ $\tiny{\pm 5.8}$ |
> > | $\texttt{tool\\_hang}$ | $14.7$ $\tiny{\pm 5.8}$ | $19.3$ $\tiny{\pm 2.3}$ | $\mathbf{38.0}$ $\tiny{\pm 7.2}$ | $14.7$ $\tiny{\pm 4.2}$ | $37.3$ $\tiny{\pm 8.3}$ | $6.7$ $\tiny{\pm 4.6}$ | $22.0$ $\tiny{\pm 8.7}$ |
> > | $\texttt{transport}$ | $23.3$ $\tiny{\pm 5.0}$ | $20.7$ $\tiny{\pm 4.2}$ | $\mathbf{32.0}$ $\tiny{\pm 7.2}$ | $8.0$ $\tiny{\pm 2.0}$ | $14.7$ $\tiny{\pm 8.3}$ | $2.7$ $\tiny{\pm 1.2}$ | $5.3$ $\tiny{\pm 2.3}$ |
> > | **average** | $58.1$ | $59.6$ | $\mathbf{65.2}$ | $54.7$ | $62.7$ | $42.5$ | $51.6$ |
> >
> >
> > The table above compares the performances from different action chunk sizes $N$ on the five Robomimic tasks (at 500K steps, 3 seeds each, $\pm$ denotes standard deviations). The results suggest that, as long as $N$ is in an appropriate range (mostly within 3-5), V-Former with open-loop control ($k = 3$) achieves the best or near-best performance consistently across the five different environments. As such, we may sweep the optimal hyperparameter $N$ in some representative tasks, and reuse the best $N$ for the other tasks. We will add the ablation results above to the paper.
> >
> >
> >
> > * **“How are the success rates in Fig 3 normalized? Why are some greater than 1?”**
> >
> > In Fig. 3, we first compute the mean success rate of the six methods (in Tables 1 and 2) for each task. We then normalize the raw performances by dividing them by the mean success rates. As a result, some normalized scores may exceed $1$, as normalization is done across different methods. The rationale behind this normalization is to ensure that tasks with high success rates do not disproportionately influence the overall results (otherwise, tasks with low success rates will almost be ignored when we naively aggregate the results). We have further clarified this in the revised manuscript.
> >
> >
> > We thank the reviewer again for the helpful feedback and please let us know if there are any additional concerns or questions.

---

> > > ### Author Response · Authors · 2023-11-21
> > > **Gentle Reminder for Reviewer Feedback**
> > >
> > > We greatly appreciate your time and dedication to providing us with your valuable feedback. We hope we have addressed the concerns, but if there is anything else that needs clarification or further discussion, please do not hesitate to let us know.

---

> > > > ### Comment · Reviewer_ZdXy · 2023-11-22
> > > >
> > > > Thanks for the response and additional experiments. My major concern on the baselines have been addressed. I've thus increased my score.

---

### Official Review · Reviewer_yZ1P · 2023-10-28

**Soundness:** 4 excellent
**Presentation:** 3 good
**Contribution:** 3 good
**Rating:** 6
**Confidence:** 3

**Summary:**

The authors present an offline RL approach built on implicit Q/V-learning, extending the original formulation to n-steps, enabling offline value learning from arbitrary-length actions. They further formulate the Bellman equation in a continuous-time MDP, therefore supporting learning over multiple datasets with distinct temporal frequencies, as is commonly available in robotics. The authors train open loop, multistep transformer policies, taking temporally extended actions, with advantage weighted regression, learning from suboptimal data, outperforming prior approaches on robotic benchmarks w/wo multiple temporal frequencies.

**Strengths:**

1) Extending implicit V-learning to n-steps is intuitive and well motivated.
2) Results are impressive, especially on increasingly suboptimal datasets.
3) Table 4 ablation study is appreciated.
4) Paper is well written and the approach should be simple to implement and adopt by the wider community.

**Weaknesses:**

1) The authors do not report confidence intervals in many of their results
2) Only 3 random seeds were ran, which is very low
3) Setting the action chunking length as a hyperparameter seems restrictive. Wouldn’t it be better to learn dynamic chunking lengths, based on the task and state? E.g., wouldn’t something more akin to “options” [1,2] work better here?

[1] - Sutton, Richard S., Doina Precup, and Satinder Singh. "Intra-Option Learning about Temporally Abstract Actions." ICML. Vol. 98. 1998.

[2] - Salter, Sasha, et al. "Mo2: Model-based offline options." Conference on Lifelong Learning Agents. PMLR, 2022.

**Questions:**

Could the authors comment on what the choice of ‘n’ in Eq 9 has during learning?

---

> ### Author Response · Authors · 2023-11-19
>
> We thank the reviewer for the thorough review and constructive feedback about this work. Below, we provide answers to the questions. We welcome additional suggestions for further improving the work.
>
> * **More random seeds and confidence intervals**
>
> Thanks for the suggestion. While we reported two aggregation plots with confidence intervals (Fig. 3), in which V-Former shows the best performance by statistically significant margins, we completely agree with the reviewer’s suggestion and will add at least three more seeds (i.e., six seeds in total) as well as statistics to the final version of the paper (for the statistics of our current results, please see our response to Reviewer 7pis). Please understand that we were not able to run all $6$ Transformer methods on $5 \times 2$ datasets with $3$ more seeds within the limited time of the rebuttal period.
>
> * **“Setting the action chunking length as a hyperparameter seems restrictive. Wouldn’t it be better to learn dynamic chunking lengths, based on the task and state?”**
>
> Thanks for raising this point. In an earlier version of this work, we made initial attempts at devising an algorithm to select action lengths dynamically based on learned advantages. Specifically, we tried adding a `STOP` token to the action space and training a policy to output `STOP` if there are no suitable open-loop actions. However, we didn’t find this version to be particularly better than the current fixed-step policy in our experiments and thus omitted it from the paper. That being said, we believe extending V-Former to handle dynamic chunk lengths is an exciting future research direction.
>
> * **Could the authors comment on what the choice of ‘n’ in Eq 9 has during learning?**
>
> As stated in L4 of Algorithm 1, we sample $n$ from the uniform distribution over $\{1, 2, \dots, N\}$. We use $N=3$ for Robomimic and $N=12$ for Kitchen.
>
>
> We thank the reviewer again for the helpful feedback and please let us know if there are any additional concerns or questions.

---

### Official Review · Reviewer_B2yt · 2023-11-01

**Soundness:** 2 fair
**Presentation:** 3 good
**Contribution:** 2 fair
**Rating:** 3
**Confidence:** 4

**Summary:**

This paper proposes V-Former, an offline RL algorithm to learn from suboptimal, multi-modal, and non-Markovian data with different control frequencies. To address these challenges, the authors first extend implicit V-learning to arbitrary frequencies, and then train a Transformer policy with advantage reweighting to produce temporally extended actions. The empirical results show that V-Former can adapt to time-heterogeneous datasets and outperform its per-timestep or BC variants.

**Strengths:**

This paper aims to address some important questions in offline RL. It is clearly written and the proposed algorithm is novel to my knowledge. The idea of using Transformer to generate temporally extended "action chunks" sounds interesting.

**Weaknesses:**

While well-motivated, I have some major concerns about the methodology and experiments of this work:

1. What value is $V_\psi(s)$ modeling in value learning? According to Equation 5-9, it seems that $V_\psi(s)$ is trying to approximate the value of the optimal single-step policy with the n-step Bellman equation. However, $V_\psi(s)$ is proposed to model the value of arbitrary action frequencies or action lengths that should have different values, which is confusing to me.
2. Compared to previous offline hierarchical RL works, what's the advantage of the proposed method? These works [1, 2, 3] also aim to solve similar tasks.
3. I am worried that the experiments are insufficient to support that V-Former is a strong baseline for offline RL, as we can only see ablations of V-Former on action chunking and advantage weighting, missing the performance of other state-of-the-art offline RL and offline HRL baselines. Moreover, Section 5.3 shows that the optimal action chunk size is around 3 in Robomimic tasks, which makes it hard to distinguish the effect of temporally extended actions. Therefore, I suggest authors compare the performance of V-Former and other baselines on tasks that may benefit from longer horizon control, such as *antmaze* and *kitchen* in D4RL.

Minor questions:

1. How to choose the hyper-parameter N? It is unclear to me the criteria for choosing N in different environments during evaluation.
2. The results in Section 5.1 and 5.3 indicate that open-loop control can achieve the best performance. However, Section 5.2 uses a close-loop VF for evaluation. Can authors provide some intuitions behind this choice?

Overall, I am unable to recommend acceptance at this stage given the questions mentioned above. However, I would consider raising my score if the authors could address my concerns.

[1] Pertsch, Karl, Youngwoon Lee, and Joseph Lim. "Accelerating reinforcement learning with learned skill priors." Conference on robot learning. PMLR, 2021.

[2] Ajay, Anurag, et al. "Opal: Offline primitive discovery for accelerating offline reinforcement learning." arXiv preprint arXiv:2010.13611 (2020).

[3] Yang, Yiqin, et al. "Flow to control: Offline reinforcement learning with lossless primitive discovery." Proceedings of the AAAI Conference on Artificial Intelligence. Vol. 37. No. 9. 2023.

**Questions:**

There are some questions and concerns, which I have outlined in the previous section.

---

> ### Author Response · Authors · 2023-11-19
>
> We thank the reviewer for the thorough review and constructive feedback about this work. Below, we describe how we have added an additional comparison with $3$ new baselines and a new ablation study with different action chunk sizes. We believe that these changes strengthen the paper, and welcome additional suggestions for further improving the work.
>
> * **$\mathbf{3}$ new baselines - CQL, IQL, and DT**
>
> Thank you for the suggestion. We would like to first note that we have (already) included comparisons with three existing offline RL and BC methods, (1) BC, (2) BC+Transformer, and (3) IQL (which correspond to $BC (1, 1)$, $BC (3, 3)$, and $VF (1, 1)$, respectively (Tables 1, 2, Fig. 3); these methods use the same action discretization as V-Former). However, following the suggestion, below, we compare V-Former with $\mathbf{3}$ additional existing offline RL methods: **CQL**, **(Original) IQL**, and **Decision Transformer**.
>
> **(1) Results on expert (PH) datasets:**
> | Task | BC | BC+Transformer | IQL | CQL | (Original) IQL | Decision Transformer | V-Former (ours) |
> | :----: | :----: | :----: | :----: | :----: | :----: | :----: | :----: |
> | $\texttt{can}$ | $90.0$ $\tiny{\pm 3.5}$ | $91.3$ $\tiny{\pm 9.0}$ | $92.7$ $\tiny{\pm 6.1}$ | $10.0$ $\tiny{\pm 11.1}$ | $37.3$ $\tiny{\pm 9.5}$ | $93.3$ $\tiny{\pm 1.2}$ | $\mathbf{95.3}$ $\tiny{\pm 3.1}$ |
> | $\texttt{lift}$ | $88.7$ $\tiny{\pm 9.5}$ | $94.7$ $\tiny{\pm 2.3}$ | $94.7$ $\tiny{\pm 4.2}$ | $70.7$ $\tiny{\pm 11.5}$ | $74.0$ $\tiny{\pm 8.7}$ | $\mathbf{98.0}$ $\tiny{\pm 2.0}$ | $93.3$ $\tiny{\pm 3.1}$ |
> | $\texttt{square}$ | $62.7$ $\tiny{\pm 5.0}$ | $\mathbf{78.0}$ $\tiny{\pm 7.2}$ | $65.3$ $\tiny{\pm 8.1}$ | $0.0$ $\tiny{\pm 0.0}$ | $31.3$ $\tiny{\pm 9.0}$ | $30.7$ $\tiny{\pm 6.4}$ | $67.3$ $\tiny{\pm 1.2}$ |
> | $\texttt{tool\\_hang}$ | $14.7$ $\tiny{\pm 8.3}$ | $36.7$ $\tiny{\pm 8.1}$ | $14.7$ $\tiny{\pm 5.8}$ | $0.0$ $\tiny{\pm 0.0}$ | $6.0$ $\tiny{\pm 4.0}$ | $4.0$ $\tiny{\pm 2.0}$ | $\mathbf{38.0}$ $\tiny{\pm 7.2}$ |
> | $\texttt{transport}$ | $22.7$ $\tiny{\pm 3.1}$ | $\mathbf{35.3}$ $\tiny{\pm 8.1}$ | $23.3$ $\tiny{\pm 5.0}$ | $0.0$ $\tiny{\pm 0.0}$ | $2.0$ $\tiny{\pm 0.0}$ | $2.0$ $\tiny{\pm 2.0}$ | $32.0$ $\tiny{\pm 7.2}$ |
> | **average** | $55.7$ | $\mathbf{67.2}$ | $58.1$ | $16.1$ | $30.1$ | $45.6$ | $65.2$ |
>
> **(2) Results on suboptimal (mixed) datasets:**
> | Task | BC | BC+Transformer | IQL | CQL | (Original) IQL | Decision Transformer | V-Former (ours) |
> | :----: | :----: | :----: | :----: | :----: | :----: | :----: | :----: |
> | $\texttt{can}$ | $52.0$ $\tiny{\pm 7.2}$ | $59.3$ $\tiny{\pm 10.3}$ | $54.7$ $\tiny{\pm 10.3}$ | $0.0$ $\tiny{\pm 0.0}$ | $10.7$ $\tiny{\pm 8.3}$ | $33.3$ $\tiny{\pm 17.5}$ | $\mathbf{66.7}$ $\tiny{\pm 5.8}$ |
> | $\texttt{lift}$ | $50.0$ $\tiny{\pm 9.2}$ | $58.7$ $\tiny{\pm 4.2}$ | $58.7$ $\tiny{\pm 9.9}$ | $0.0$ $\tiny{\pm 0.0}$ | $26.0$ $\tiny{\pm 4.0}$ | $60.0$ $\tiny{\pm 3.5}$ | $\mathbf{70.0}$ $\tiny{\pm 15.6}$ |
> | $\texttt{square}$ | $37.3$ $\tiny{\pm 9.5}$ | $\mathbf{66.0}$ $\tiny{\pm 8.0}$ | $47.3$ $\tiny{\pm 5.8}$ | $0.0$ $\tiny{\pm 0.0}$ | $18.0$ $\tiny{\pm 6.0}$ | $6.0$ $\tiny{\pm 7.2}$ | $\mathbf{66.0}$ $\tiny{\pm 3.5}$ |
> | $\texttt{tool\\_hang}$ | $11.3$ $\tiny{\pm 5.0}$ | $28.0$ $\tiny{\pm 14.4}$ | $15.3$ $\tiny{\pm 4.2}$ | $0.0$ $\tiny{\pm 0.0}$ | $0.0$ $\tiny{\pm 0.0}$ | $0.0$ $\tiny{\pm 0.0}$ | $\mathbf{28.7}$ $\tiny{\pm 7.0}$ |
> | $\texttt{transport}$ | $8.0$ $\tiny{\pm 2.0}$ | $16.0$ $\tiny{\pm 6.9}$ | $14.7$ $\tiny{\pm 5.0}$ | $0.0$ $\tiny{\pm 0.0}$ | $0.0$ $\tiny{\pm 0.0}$ | $0.0$ $\tiny{\pm 0.0}$ | $\mathbf{25.3}$ $\tiny{\pm 2.3}$ |
> | **average** | $31.7$ | $45.6$ | $38.1$ | $0.0$ | $10.9$ | $19.9$ | $\mathbf{51.3}$ |
>
> The table above shows the comparison results of V-Former with three additional baselines on both expert and mixed (suboptimal) datasets (at 500K steps, 3 seeds each, $\pm$ denotes standard deviations). The results suggest that V-Former mostly outperforms the three new baselines, especially on challenging mixed datasets. In particular, previous offline RL methods that do not use temporally extended actions (i.e., CQL and IQL) struggle on these narrow, suboptimal datasets since their policies cannot fully represent highly non-Markovian and multi-modal behavioral policies. We will add these results to the paper.

---

> > ### Author Response · Authors · 2023-11-19
> >
> > * **“What value is V(s) modeling in value learning?”**
> >
> > Our $V_\psi(s)$ approximates $V^*(s)$, the *optimal* value function of the MDP. As the reviewer noted, this is exactly the same as the optimal value function of a *single-step* policy. Please note that the optimal value of a multi-step or arbitrary-length policy is always no larger than that of a single-step policy in deterministic environments, and thus it is always “safe” to get bootstrap updates from multi-step returns, as in Eq. 8 or 9. As a result, in theory, both the one-step IVL update (Eq. 4) and our multi-step IVL update (Eq. 9) converge to the same optimum (when $\tau \to 1$). However, in practice, this multi-step generalization can often be beneficial because it expedites value learning and allows us to utilize time-heterogeneous data.
> >
> > * **How to select the action chunk size $N$?**
> >
> > As the reviewer pointed out, the action chunk size $N$ is a hyperparameter that we need to tune, as in most previous works in hierarchical RL and multi-step BC. However, we found that the optimal action chunk size $N$ is *not* very sensitive to individual tasks. We present the ablation results of V-Former (VF) on the five environments from Robomimic below:
> >
> > | Method ($N$, $k$) | VF (1, 1) | VF (3, 1) | VF (3, 3) | VF (5, 1) | VF (5, 3) | VF (8, 1) | VF (8, 3) |
> > | :----: | :----: | :----: | :----: | :----: | :----: | :----: | :----: |
> > | $\texttt{can}$ | $92.7$ $\tiny{\pm 6.1}$ | $90.7$ $\tiny{\pm 3.1}$ | $\mathbf{95.3}$ $\tiny{\pm 3.1}$ | $91.3$ $\tiny{\pm 4.2}$ | $91.3$ $\tiny{\pm 2.3}$ | $92.0$ $\tiny{\pm 3.5}$ | $94.7$ $\tiny{\pm 2.3}$ |
> > | $\texttt{lift}$ | $\mathbf{94.7}$ $\tiny{\pm 4.2}$ | $94.0$ $\tiny{\pm 2.0}$ | $93.3$ $\tiny{\pm 3.1}$ | $93.3$ $\tiny{\pm 3.1}$ | $\mathbf{94.7}$ $\tiny{\pm 1.2}$ | $43.3$ $\tiny{\pm 8.3}$ | $71.3$ $\tiny{\pm 8.1}$ |
> > | $\texttt{square}$ | $65.3$ $\tiny{\pm 8.1}$ | $73.3$ $\tiny{\pm 3.1}$ | $67.3$ $\tiny{\pm 1.2}$ | $66.0$ $\tiny{\pm 8.7}$ | $\mathbf{75.3}$ $\tiny{\pm 6.1}$ | $68.0$ $\tiny{\pm 13.1}$ | $64.7$ $\tiny{\pm 5.8}$ |
> > | $\texttt{tool\\_hang}$ | $14.7$ $\tiny{\pm 5.8}$ | $19.3$ $\tiny{\pm 2.3}$ | $\mathbf{38.0}$ $\tiny{\pm 7.2}$ | $14.7$ $\tiny{\pm 4.2}$ | $37.3$ $\tiny{\pm 8.3}$ | $6.7$ $\tiny{\pm 4.6}$ | $22.0$ $\tiny{\pm 8.7}$ |
> > | $\texttt{transport}$ | $23.3$ $\tiny{\pm 5.0}$ | $20.7$ $\tiny{\pm 4.2}$ | $\mathbf{32.0}$ $\tiny{\pm 7.2}$ | $8.0$ $\tiny{\pm 2.0}$ | $14.7$ $\tiny{\pm 8.3}$ | $2.7$ $\tiny{\pm 1.2}$ | $5.3$ $\tiny{\pm 2.3}$ |
> > | **average** | $58.1$ | $59.6$ | $\mathbf{65.2}$ | $54.7$ | $62.7$ | $42.5$ | $51.6$ |
> >
> >
> > The table above compares the performances from different action chunk sizes $N$ on the five Robomimic tasks (at 500K steps, 3 seeds each, $\pm$ denotes standard deviations). The results suggest that, as long as $N$ is in an appropriate range (mostly within 3-5), V-Former with open-loop control ($k = 3$) achieves the best or near-best performance consistently across the five different environments. As such, we may sweep the optimal hyperparameter $N$ in some representative tasks, and reuse the best $N$ for the other tasks. We will add the ablation results above to the paper.
> >
> > * **Open-loop vs. closed-loop action execution**
> >
> > Thank you for raising this question. In general, we found that open-loop control is often better than closed-loop control when the behavioral policies are highly non-Markovian, since closed-loop actions (at evaluation time) cannot simulate a non-Markovian policy. This is also observed by Zhao et al. [1]. Otherwise, closed-loop control is often better than open-loop control because it can be more reactive to observations. In our environments, Robomimic datasets consist of highly non-Markovian human demonstrations [2], while the time-heterogeneous Kitchen dataset mostly consists of trajectories generated by a Markovian policy. As such, we use open-loop control for Robomimic and closed-loop control for Kitchen. We have clarified this in the revised manuscript.
> >
> > We thank the reviewer again for the helpful feedback and please let us know if there are any additional concerns or questions.
> >
> > [1] Zhao et al., Learning Fine-Grained Bimanual Manipulation with Low-Cost Hardware (2023). \
> > [2] Mandlekar et al., What Matters in Learning from Ofﬂine Human Demonstrations for Robot Manipulation (2021).

---

> > > ### Author Response · Authors · 2023-11-21
> > > **Gentle Reminder for Reviewer Feedback**
> > >
> > > We greatly appreciate your time and dedication to providing us with your valuable feedback. We hope we have addressed the concerns, but if there is anything else that needs clarification or further discussion, please do not hesitate to let us know.

---

> > > ### Comment · Reviewer_B2yt · 2023-11-21
> > >
> > > Thank you for your clarification and additional experiments. While the comparison with new baselines demonstrates V-Former's better performance on Robomimic tasks, my major concerns may not be fully addressed in the response. Here are my further questions:
> > >
> > > * **Comparison to new baselines.** I appreciate these results presented by authors, which show that V-Former is better than some state-of-the-art offline RL algorithms in some cases. However, comparing the performance of IQL(VF(1,1) in the paper) and IQL(Original), I feel that it is the Transformer rather than the temporal abstraction that contributes to the most performance gain. Moreover, given that original IQL, CQL, and DT are not evaluated on Robomimic tasks, I think it is more fair to compare these methods on the standard offline RL benchmark such as D4RL.
> > > * **Main contribution of this work.** The authors claim that the significance of this work is "the effectiveness of temporally extended actions in the offline RL setting" in their response to Reviewer 7pis. However, such an idea has been explored in many offline hierarchical RL methods (see my review part point 2 and citation [1, 2, 3]). It seems that neither theoretical justification nor empirical results could show that V-Former is better than these offline hierarchical RL methods.
> > > * **The action chunk size $N$.** I agree that the table reported can show that $N$ is not sensitive to 5 tasks in Robomimic, possibly due to the agent controlling the same robot arm. However, according to Table 3 on Kitchen environment, the optimal $N$ may vary across different robots, which may need us to sweep the hyperparameter.
> > > * **Open-loop vs. closed-loop action execution.** I appreciate this discussion. However, it is unclear to me why closed-loop actions (at evaluation time) cannot simulate a non-Markovian policy. For instance, you can model the low-level policy as DT, which is non-Markovian but can perform closed-loop control.

---

### Official Review · Reviewer_7pis · 2023-11-07

**Soundness:** 2 fair
**Presentation:** 2 fair
**Contribution:** 2 fair
**Rating:** 3
**Confidence:** 4

**Summary:**

This paper focuses on handling data with different qualities (suboptimal) and different frequencies. The paper proposes a method, V-former, which utilizes the idea of “action trunks” and a transformer-based policy. Concretely, it extends the value function of Implicit Value Learning to bootstrap with multiple steps actions and uses the transformer policy to roll out multiple steps. In five robomimic tasks with different data qualities and kitchen tasks with different data frequencies, the proposed method shows better performance than baselines.

**Strengths:**

- The paper is well-organized and clear.

 - The method section is easy to follow.

**Weaknesses:**

- The method is straightforward, and the contribution is limited. The main technical contribution of the paper is extending the value function of IVL and making it consider the outcome of multiple timesteps, which I believe is not significant enough. The underlying insight that modeling multiple steps to help handle multimodality is already known in the literature.

 - The experiment evaluation is not thorough enough. The baselines are mostly ablation of the proposed method. Moreover, there are other existing offline RL methods that also can be applied to the problem of interest, such as IQL. The limited set of experiments makes the significance of the proposed method hard to evaluate.

**Questions:**

- The evaluation in Table 1 is interesting and shows that VF can achieve good performance if proper N and k are selected. However, the optimal N and k may be quite different for different tasks. Instead of manually selecting them, will there be a general way to derive them from the offline dataset?

---

> ### Author Response · Authors · 2023-11-19
>
> We thank the reviewer for the thorough review and constructive feedback about this work. Below, we describe how we have added an additional comparison with $3$ existing offline RL methods and a new ablation study with different action chunk sizes. We believe that these changes strengthen the paper, and welcome additional suggestions for further improving the work.
>
> * **$\mathbf{3}$ new baselines - CQL, IQL, and DT**
>
> Thank you for the suggestion. We would like to first note that we have (already) included comparisons with three existing offline RL and BC methods, (1) BC, (2) BC+Transformer, and (3) IQL (which correspond to $BC (1, 1)$, $BC (3, 3)$, and $VF (1, 1)$, respectively (Tables 1, 2, Fig. 3); these methods use the same action discretization as V-Former). However, following the suggestion, below, we compare V-Former with $\mathbf{3}$ additional existing offline RL methods: **CQL**, **(Original) IQL**, and **Decision Transformer**.
>
> **(1) Results on expert (PH) datasets:**
> | Task | BC | BC+Transformer | IQL | CQL | (Original) IQL | Decision Transformer | V-Former (ours) |
> | :----: | :----: | :----: | :----: | :----: | :----: | :----: | :----: |
> | $\texttt{can}$ | $90.0$ $\tiny{\pm 3.5}$ | $91.3$ $\tiny{\pm 9.0}$ | $92.7$ $\tiny{\pm 6.1}$ | $10.0$ $\tiny{\pm 11.1}$ | $37.3$ $\tiny{\pm 9.5}$ | $93.3$ $\tiny{\pm 1.2}$ | $\mathbf{95.3}$ $\tiny{\pm 3.1}$ |
> | $\texttt{lift}$ | $88.7$ $\tiny{\pm 9.5}$ | $94.7$ $\tiny{\pm 2.3}$ | $94.7$ $\tiny{\pm 4.2}$ | $70.7$ $\tiny{\pm 11.5}$ | $74.0$ $\tiny{\pm 8.7}$ | $\mathbf{98.0}$ $\tiny{\pm 2.0}$ | $93.3$ $\tiny{\pm 3.1}$ |
> | $\texttt{square}$ | $62.7$ $\tiny{\pm 5.0}$ | $\mathbf{78.0}$ $\tiny{\pm 7.2}$ | $65.3$ $\tiny{\pm 8.1}$ | $0.0$ $\tiny{\pm 0.0}$ | $31.3$ $\tiny{\pm 9.0}$ | $30.7$ $\tiny{\pm 6.4}$ | $67.3$ $\tiny{\pm 1.2}$ |
> | $\texttt{tool\\_hang}$ | $14.7$ $\tiny{\pm 8.3}$ | $36.7$ $\tiny{\pm 8.1}$ | $14.7$ $\tiny{\pm 5.8}$ | $0.0$ $\tiny{\pm 0.0}$ | $6.0$ $\tiny{\pm 4.0}$ | $4.0$ $\tiny{\pm 2.0}$ | $\mathbf{38.0}$ $\tiny{\pm 7.2}$ |
> | $\texttt{transport}$ | $22.7$ $\tiny{\pm 3.1}$ | $\mathbf{35.3}$ $\tiny{\pm 8.1}$ | $23.3$ $\tiny{\pm 5.0}$ | $0.0$ $\tiny{\pm 0.0}$ | $2.0$ $\tiny{\pm 0.0}$ | $2.0$ $\tiny{\pm 2.0}$ | $32.0$ $\tiny{\pm 7.2}$ |
> | **average** | $55.7$ | $\mathbf{67.2}$ | $58.1$ | $16.1$ | $30.1$ | $45.6$ | $65.2$ |
>
> **(2) Results on suboptimal (mixed) datasets:**
> | Task | BC | BC+Transformer | IQL | CQL | (Original) IQL | Decision Transformer | V-Former (ours) |
> | :----: | :----: | :----: | :----: | :----: | :----: | :----: | :----: |
> | $\texttt{can}$ | $52.0$ $\tiny{\pm 7.2}$ | $59.3$ $\tiny{\pm 10.3}$ | $54.7$ $\tiny{\pm 10.3}$ | $0.0$ $\tiny{\pm 0.0}$ | $10.7$ $\tiny{\pm 8.3}$ | $33.3$ $\tiny{\pm 17.5}$ | $\mathbf{66.7}$ $\tiny{\pm 5.8}$ |
> | $\texttt{lift}$ | $50.0$ $\tiny{\pm 9.2}$ | $58.7$ $\tiny{\pm 4.2}$ | $58.7$ $\tiny{\pm 9.9}$ | $0.0$ $\tiny{\pm 0.0}$ | $26.0$ $\tiny{\pm 4.0}$ | $60.0$ $\tiny{\pm 3.5}$ | $\mathbf{70.0}$ $\tiny{\pm 15.6}$ |
> | $\texttt{square}$ | $37.3$ $\tiny{\pm 9.5}$ | $\mathbf{66.0}$ $\tiny{\pm 8.0}$ | $47.3$ $\tiny{\pm 5.8}$ | $0.0$ $\tiny{\pm 0.0}$ | $18.0$ $\tiny{\pm 6.0}$ | $6.0$ $\tiny{\pm 7.2}$ | $\mathbf{66.0}$ $\tiny{\pm 3.5}$ |
> | $\texttt{tool\\_hang}$ | $11.3$ $\tiny{\pm 5.0}$ | $28.0$ $\tiny{\pm 14.4}$ | $15.3$ $\tiny{\pm 4.2}$ | $0.0$ $\tiny{\pm 0.0}$ | $0.0$ $\tiny{\pm 0.0}$ | $0.0$ $\tiny{\pm 0.0}$ | $\mathbf{28.7}$ $\tiny{\pm 7.0}$ |
> | $\texttt{transport}$ | $8.0$ $\tiny{\pm 2.0}$ | $16.0$ $\tiny{\pm 6.9}$ | $14.7$ $\tiny{\pm 5.0}$ | $0.0$ $\tiny{\pm 0.0}$ | $0.0$ $\tiny{\pm 0.0}$ | $0.0$ $\tiny{\pm 0.0}$ | $\mathbf{25.3}$ $\tiny{\pm 2.3}$ |
> | **average** | $31.7$ | $45.6$ | $38.1$ | $0.0$ | $10.9$ | $19.9$ | $\mathbf{51.3}$ |
>
> The table above shows the comparison results of V-Former with three additional baselines on both expert and mixed (suboptimal) datasets (at 500K steps, 3 seeds each, $\pm$ denotes standard deviations). The results suggest that V-Former mostly outperforms the three new baselines, especially on challenging mixed datasets. In particular, previous offline RL methods that do not use temporally extended actions (i.e., CQL and IQL) struggle on these narrow, suboptimal datasets since their policies cannot fully represent highly non-Markovian and multi-modal behavioral policies. We will add these results to the paper.

---

> ### Author Response · Authors · 2023-11-19
>
> * **“The method is straightforward, and the contribution is limited”**, **“The underlying insight that modeling multiple steps to help handle multimodality is already known in the literature”**
>
> The significance of our work is that we demonstrate the effectiveness of temporally extended actions in the **offline RL** setting, especially when the dataset consists of narrow demonstration data and broader suboptimal data, a setting very common in the real world (e.g., large task-agnostic + small task-specific demonstrations). While V-Former is indeed straightforward to implement, we believe that this simplicity is a strength of the approach, not a weakness. Also, we are not aware of any prior work that uses such a multi-step open-loop policy for offline RL. Applying the idea of action chunking to offline RL is not straightforward as it requires fitting a multi-step Q function, which we resolve by generalizing an in-sample value maximization algorithm in a novel way. Finally, we would like to note that the V-Former can even handle **time-heterogeneous** datasets thanks to our generalized IVL, outperforming previous work (Burns et al., 2022) by a significant margin (Table 3).
>
>
> * **How to select the action chunk size $N$?**
>
> As the reviewer pointed out, the action chunk size $N$ is a hyperparameter that we need to tune, as in most previous works in hierarchical RL and multi-step BC. However, we found that the optimal action chunk size $N$ is *not* very sensitive to individual tasks. We present the ablation results of V-Former (VF) on the five environments from Robomimic below:
>
> | Method ($N$, $k$) | VF (1, 1) | VF (3, 1) | VF (3, 3) | VF (5, 1) | VF (5, 3) | VF (8, 1) | VF (8, 3) |
> | :----: | :----: | :----: | :----: | :----: | :----: | :----: | :----: |
> | $\texttt{can}$ | $92.7$ $\tiny{\pm 6.1}$ | $90.7$ $\tiny{\pm 3.1}$ | $\mathbf{95.3}$ $\tiny{\pm 3.1}$ | $91.3$ $\tiny{\pm 4.2}$ | $91.3$ $\tiny{\pm 2.3}$ | $92.0$ $\tiny{\pm 3.5}$ | $94.7$ $\tiny{\pm 2.3}$ |
> | $\texttt{lift}$ | $\mathbf{94.7}$ $\tiny{\pm 4.2}$ | $94.0$ $\tiny{\pm 2.0}$ | $93.3$ $\tiny{\pm 3.1}$ | $93.3$ $\tiny{\pm 3.1}$ | $\mathbf{94.7}$ $\tiny{\pm 1.2}$ | $43.3$ $\tiny{\pm 8.3}$ | $71.3$ $\tiny{\pm 8.1}$ |
> | $\texttt{square}$ | $65.3$ $\tiny{\pm 8.1}$ | $73.3$ $\tiny{\pm 3.1}$ | $67.3$ $\tiny{\pm 1.2}$ | $66.0$ $\tiny{\pm 8.7}$ | $\mathbf{75.3}$ $\tiny{\pm 6.1}$ | $68.0$ $\tiny{\pm 13.1}$ | $64.7$ $\tiny{\pm 5.8}$ |
> | $\texttt{tool\\_hang}$ | $14.7$ $\tiny{\pm 5.8}$ | $19.3$ $\tiny{\pm 2.3}$ | $\mathbf{38.0}$ $\tiny{\pm 7.2}$ | $14.7$ $\tiny{\pm 4.2}$ | $37.3$ $\tiny{\pm 8.3}$ | $6.7$ $\tiny{\pm 4.6}$ | $22.0$ $\tiny{\pm 8.7}$ |
> | $\texttt{transport}$ | $23.3$ $\tiny{\pm 5.0}$ | $20.7$ $\tiny{\pm 4.2}$ | $\mathbf{32.0}$ $\tiny{\pm 7.2}$ | $8.0$ $\tiny{\pm 2.0}$ | $14.7$ $\tiny{\pm 8.3}$ | $2.7$ $\tiny{\pm 1.2}$ | $5.3$ $\tiny{\pm 2.3}$ |
> | **average** | $58.1$ | $59.6$ | $\mathbf{65.2}$ | $54.7$ | $62.7$ | $42.5$ | $51.6$ |
>
>
> The table above compares the performances from different action chunk sizes $N$ on the five Robomimic tasks (at 500K steps, 3 seeds each, $\pm$ denotes standard deviations). The results suggest that, as long as $N$ is in an appropriate range (mostly within 3-5), V-Former with open-loop control ($k = 3$) achieves the best or near-best performance consistently across the five different environments. As such, we may sweep the optimal hyperparameter $N$ in some representative tasks, and reuse the best $N$ for the other tasks. We will add the ablation results above to the paper.
>
> We thank the reviewer again for the helpful feedback and please let us know if there are any additional concerns or questions.

---

> > ### Author Response · Authors · 2023-11-21
> > **Gentle Reminder for Reviewer Feedback**
> >
> > We greatly appreciate your time and dedication to providing us with your valuable feedback. We hope we have addressed the concerns, but if there is anything else that needs clarification or further discussion, please do not hesitate to let us know.

---

### Meta-Review · Area_Chair_ysec · 2023-12-11

**Metareview:**

This paper proposes V-former, a method tailored for offline RL with temporally extended actions, which addresses challenges of mixed demonstration and suboptimal data, as well as control frequency issues. The method extends the implicit Q-learning (IQL) approach by incorporating temporally extended "action chunks" and Transformer-based policies to efficiently perform offline RL with continuous time formulation. This combination allows for effective representation of open-loop action sequences in robotic settings. There are some weaknesses of the paper raised from the review comments and discussions, including the incremental technical novelty (or advantages), insufficient experiments. Although the authors provided detailed feedbacks, some of the concerns raised are still unsolved.

**Justification For Why Not Higher Score:**

There are some weaknesses of the paper raised from the review comments and discussions, including the incremental technical novelty (or advantages), insufficient experiments. Although the authors provided detailed feedbacks, some of the concerns raised are still unsolved.

**Justification For Why Not Lower Score:**

N/A

---

### Decision · Program_Chairs · 2024-01-16

Reject